# In Vitro Production and Immunogenicity of a *Clostridium difficile* Spore-Specific BclA3 Glycopeptide Conjugate Vaccine

**DOI:** 10.3390/vaccines8010073

**Published:** 2020-02-07

**Authors:** Annie Aubry, Wei Zou, Evguenii Vinogradov, Dean Williams, Wangxue Chen, Greg Harris, Hongyan Zhou, Melissa J. Schur, Michel Gilbert, Gillian R. Douce, Susan M. Logan

**Affiliations:** 1Vaccine Program, Human Health Therapeutics Research Centre, National Research Council of Canada, Ottawa, ON K1A 0R6, Canada; annie.aubry@nrc-cnrc.gc.ca (A.A.); wei.zou@nrc-cnrc.gc.ca (W.Z.); evguenii.vinogradov@nrc-cnrc.gc.ca (E.V.); dean.williams@nrc-cnrc.gc.ca (D.W.); Wangxue.Chen@nrc-cnrc.gc.ca (W.C.); Greg.harris@nrc-cnrc.gc.ca (G.H.); Hongyan.Zhou@nrc-cnrc.gc.ca (H.Z.); Melissa.schur@nrc-cnrc.gc.ca (M.J.S.); Michel.gilbert@nrc-cnrc.gc.ca (M.G.); 2Institute of Infection, Immunity, Inflammation, College of Medical, Veterinary and Life Sciences, University of Glasgow, Glasgow G12 8TA, Scotland, UK; Gillian.Douce@glasgow.ac.uk

**Keywords:** *C. difficile* vaccine, BclA3 spore glycoprotein, *O*-GlcNAc transferase, glycopeptide, conjugate, immunogenicity

## Abstract

The BclA3 glycoprotein is a major component of the exosporangial layer of *Clostridium difficile* spores and in this study we demonstrate that this glycoprotein is a major spore surface associated antigen. Here, we confirm the role of SgtA glycosyltransferase (SgtA GT) in BclA3 glycosylation and recapitulate this process by expressing and purifying SgtA GT fused to MalE, the maltose binding protein from *Escherichia coli*. In vitro assays using the recombinant enzyme and BclA3 synthetic peptides demonstrated that SgtA GT was responsible for the addition of β-*O*-linked GlcNAc to threonine residues of each synthetic peptide. These peptide sequences were selected from the central, collagen repeat region of the BclA3 protein. Following optimization of SgtA GT activity, we generated sufficient glycopeptide (10 mg) to allow conjugation to KLH (keyhole limpet hemocyanin) protein. Glycosylated and unglycosylated versions of these conjugates were then used as antigens to immunize rabbits and mice. Immune responses to each of the conjugates were examined by Enzyme Linked Immunosorbent Assay ELISA. Additionally, the BclA3 conjugated peptide and glycopeptide were used as antigens in an ELISA assay with serum raised against formalin-killed spores. Only the glycopeptide was recognized by anti-spore polyclonal immune serum demonstrating that the glycan moiety is a predominant spore-associated surface antigen. To determine whether antibodies to these peptides could modify persistence of spores within the gut, animals immunized intranasally with either the KLH-glycopeptide or KLH-peptide conjugate in the presence of cholera toxin, were challenged with R20291 spores. Although specific antibodies were raised to both antigens, immunization did not provide any protection against acute or recurrent disease.

## 1. Introduction

*Clostridioides* (*Clostridium*) *difficile* a spore forming, anaerobic, Gram-positive organism is the major cause of antibiotic associated diarrhea. In recent years increased incidence in the number of infections, as well as increased morbidity and mortality associated with infection by *C. difficile*, has been attributed to emergence of the hypervirulent strains such as the Nap1/B1/027 ribotype [1]. Further, the rate of recurrent disease has risen dramatically with 25%, 40%, and 65% of patients relapsing after a first, second, and third episode of *C. difficile* infection (CDI), respectively, presenting a significant clinical challenge [2]. Production of spores has been shown to be a main contributing factor in CDI being essential for transmission of disease to new, susceptible patients as well as for persistence/recurrence within infected patients [3,4]. 

The process of spore formation and structural characterization is well described for several Gram-positive organisms, most notably for the human pathogen *Bacillus anthracis* [5,6]. In contrast, the spores of *C. difficile* have been less well characterized although recent publications have contributed significantly to the body of knowledge on these distinct and unique entities [7,8,9,10,11]. Whilst the regulation of sporulation and germination appears different from the classical pathways established in Bacillus [12,13], and analysis of the spore cortex demonstrates a unique and distinct proteome composition [9,10], the electron dense exosporium of *C. difficile* spores appears similar to other Gram-positive bacterial spores [14,15]. Characterization of the *C. difficile* exosporangial proteome has revealed a number of spore surface proteins, including BclA3. Previously, we showed this protein localized to an extractable, high molecular weight complex from *C. difficile* spore preparations, which could be identified in denaturing SDS-PAGE gels. Extensive in-gel proteolytic digestion of these high molecular weight complexes followed by tandem mass spectrometry analysis of the products identified a number of BclA3 peptides which were glycosylated with either a single or multiple *N*-acetyl hexosamine moieties which could be capped with a novel glycan. A glycosyltransferase gene, *sgtA*, located immediately upstream of the *bclA3* gene was shown to be involved in the glycosylation process. Inactivation of this gene led to a loss of anti-GlcNAc recognition on spore surface by immunofluorescence [14]. 

To date, vaccine development for the prevention of CDI has primarily focused on the toxins Toxin A TcdA and Toxin B TcdB produced by vegetative cells during the infection process [16,17,18]. However, attention has more recently been directed toward the spore of *C. difficile* as it is the primary agent of transmission and persistence within the gut [19]. To determine the significance of BclA3 in pathogenesis, we expressed and purified recombinant SgtA glycosyltransferase to allow in vitro synthesis of the BclA3 glycopeptide. In this paper we examine the immunogenicity of the recombinant peptide and corresponding glycopeptide after conjugation to KLH carrier protein and consider its potential to limit spore associated disease transmission in vivo.

## 2. Materials and Methods 

### 2.1. Strains 

*C. difficile* strains 630 and R20291 (provided by B. Wren LSHTM, UK) and R20291ΔsgtA [14] were routinely grown under anaerobic conditions in Don Whitely Anaerobic chamber on brain heart infusion agar medium (BD Sparks, MD, USA) supplemented with 5 g/L yeast extract, 1.2 g/L NaCl, 0.5 g/L cysteine HCl, 5 mg/L hemin, 1 mg/L vitamin K. R20291ΔsgtA was produced using Clostron mutagenesis as described by Cartman and Minton [20,21] and was grown as above with 2.5 µg/mL erythromycin.

### 2.2. Production of Spores

*C. difficile* R20291 and *C. difficile* R20291ΔsgtA cells from a mid-logarithmically grown vegetative broth culture was spread on BHIS agar plates and incubated under anaerobic conditions. Seven days later, growth was harvested into sterile distilled water and spores collected by centrifugation and extensive washing with distilled water. Spore numbers (Colony forming units (CFU)/mL) were quantified by serial dilution and plating on BHI containing 1% taurocholate (Sigma-Aldrich, St Louis, MO, USA). 

### 2.3. Recombinant Expression and Purification of SgtA 

The *sgtA* gene was cloned from genomic DNA of *C. difficile* 630 by PCR using primers sgtA-1F (GAAGCTTGAATTCATGATTACAATAAGTTTGTGCATGATTG) and sgtA-1R (GGACGCGTCGACCTACTAACTATTTTTAAATTTACTAAAATAATTTTCATTGTGC). The purified PCR product was digested with EcoR1/Sal1 and cloned into the EcoR1/Sal1 restriction sites of pCW-MalET to produce a fusion with MalE on the N terminal end of the SgtA enzyme [22]. This engineered construct was then transformed into *E. coli* AD202 cells and the transformed cells grown in 500 mL of 2xYT. Recombinant protein was induced in these cultures using 0.5 mM isopropyl-β-D-thiogalactopyranoside, 100 mg/L of ampicillin and 0.2% glucose for 16 h at room temperature. Bacterial cells were harvested by centrifugation and the cell pellet frozen at −20 ℃. Cell pellet was then resuspended in 20 mL of ice cold 20 mM Tris pH 7.5, 200 mM NaCl_2_, 1 mM EDTA buffer with 1× complete protease inhibitor cocktail tablet (Roche, Mississauga, ON, Canada ), and the suspensions disrupted with an Emulsiflex C5 instrument (Avestin, Ottawa, Ontario, Canada). After cell disruption, the debris was pelleted at 15,000× *g* for 30 min at 4 ℃ and the supernatant collected. Recombinant protein was purified by affinity chromatography on 3 mL amylose resin (New England Biolabs, Whitby, ON, Canada). After sample application, the column was washed with 30 mL of 20 mM Tris pH 7.5, 200 mM NaCl, 1 mM EDTA buffer to elute unbound proteins. The bound protein was eluted by washing the column with the same buffer containing 10 mM maltose and the eluate collected in 2 mL fractions. The fractions were analyzed by SDS-PAGE and fractions containing MalE-SgtA fusion protein were pooled and stored at 4 ℃.

### 2.4. SgtA Glycosyltransferase Assay

To examine the activity of MalE-SgtA fusion protein, assays were initially set up with synthetic peptides SML1, SML2, and SML3 which were labelled with FITC (Peptide 2.0 Chantilly, VA, USA). Additional synthetic peptides were utilized in optimized reactions. The sequence of each peptide is presented in Table 1. Addition of GlcNAc residue(s) to the peptides was detected by shift in migration on thin layer chromatography (TLC) plates developed with ethyl acetate, methanol, H_2_O, acetic acid (3:2:1:0.1), and detection using long-wave UV epi-illumination. For quantitative product determination, capillary electrophoresis (CE) analyses were performed using a P/ACE MDQ system equipped with a Laser module 488 (Beckman Coulter, Fullerton, CA, USA). Capillaries were bare silica 50 µm × 60.2 cm with a detector at 50 cm and the running buffer was 90 mM Tris/89 mM borate/2 mM EDTA, pH 8.8. Samples were introduced by pressure injection for 10 s and were run at 30 kV for 15 min. Quantification was performed by integration of trace peaks using the MDQ 32 Karat software (Beckman Coulter, Fullerton, CA, USA). Following initial analysis using 50 mM HEPES pH 7.0, 5 mM MnCl_2_/MgCl_2_, 0.2 mM FITC-Ahx-peptide (SML1, 2, 3) 1 mM UDP-GlcNAc in 10 µL reaction volume with 2 µL purified SgtA-MalE enzyme, we determined the optimal conditions for activity. Optimized buffer conditions for in vitro activity of SgtA-MalE were determined to be 50 mM HEPES pH 7.5, 10 mM MnCl_2_, 0.5 mM DTT, 0.2 mM peptide, 1 mM UDP-GlcNAc. These conditions were used in scale up reactions to produce glycopeptide for KLH conjugation. 

### 2.5. BclA3 Peptide and Glycopeptide Conjugate Production

To produce glycopeptide for conjugation, 10 mg aliquots of peptide SML10 (Peptide 2.0, Chantilly, Va, USA) was used in a scaled up in vitro assay with approx. 9 mg of purified MalE-SgtA fusion protein in assay buffer (50 mM HEPES pH 7.5, 10 mM MnCl_2_, 0.5 mM DTT, 5 mM UDP-GlcNAc) at 37 ℃ for 24 h. The SML10 glycopeptide product was purified from the reaction mixture on a Luna C18 (250 × 10 mm) reverse phase column using a water-acetonitrile gradient with 0.1% TFA as eluent (at 4 mL/min flow rate and 0%–70% for 25 min). 

### 2.6. NMR Spectroscopy 

Nuclear magnetic resonance NMR experiments were carried out on a Bruker AVANCE III 600 MHz (^1^H) spectrometer with 5 mm Z-gradient probe with acetone internal reference (2.225 ppm for ^1^H and 31.45 ppm for ^13^C) using standard pulse sequences cosygpprqf (gCOSY), mlevphpr (TOCSY, mixing time 120 ms), roesyphpr (ROESY, mixing time 500 ms), hsqcedetgp (HSQC), hsqcetgpml (HSQC-TOCSY, 80 ms TOCSY delay), and hmbcgplpndqf (HMBC, 100 ms long range transfer delay). Resolution was kept <3 Hz/pt in F2 in proton-proton correlations and <5 Hz/pt in F2 of H-C correlations. The spectra were processed and analyzed using the Bruker Topspin 2.1 program.

### 2.7. KLH-Conjugate Production

To a solution of purified SML10 glycopeptide or peptide (5 mg) in PBS (1 mL), Tris (2-carboxyethyl) phosphine HCl (TCEP) was added (1 mg in 0.1 mL PBS), and the mixture was kept at room temperature for 30 min. High Pressure Liquid chromatography HPLC analysis on a Luna C18 column was used to confirm complete reversal of the peptide dimer to monomer by reduction of the disulfide bond. Both the bovine serum albumin BSA and the keyhole limpet haemocyanin KLH were made in parallel. Each peptide sample was mixed with an equivalent amount of bromoacetylated KLH (KLH-Br) (5 mg, 2.5 mL), or, BSA-Br (0.8 mg, 0.4 mL) in PBS. The solution was kept at room temperature for 72 h. Purification of the conjugates was achieved using a 30K Centriprep or Amicon at 3000 rpm for 10 min (washed with phosphate buffered saline PBS × 5) to remove free peptides and other reagents. MALDI-MS analysis of respective BSA-conjugated peptide was used to estimate the antigen loading on the KLH conjugate, prepared in parallel, at 12%–14% (w/w). 

### 2.8. Biotin-Peptide and Biotin-Glycopeptide Production

Biotinylated SML10 peptide and SML10 glycopeptide coating antigens were prepared for ELISA analysis in a similar manner. For SML10 glycopeptide, a 2 mg solution of glycopeptide in PBS (1 mL) was mixed with TCEP (0.6 mg in 0.06 mL PBS), and the solution was kept at room temperature for 30 min when HPLC indicated complete reduction of the disulfide. Biotin-dPEG3-Mal (Sigma-Aldrich) (2 mg) in DMSO (0.1 mL) was added to this solution, and the mixture was kept at room temperature for 16 h, diluted with water, and lyophilized. Purification by HPLC on a Luna C18 column (250 × 10 mm) afforded the glycopeptide-biotin conjugate which was confirmed by ES-MS as white solids after lyophilization (MonoHexNAc-Glycopeptide-Biotin (2.8 mg), ES MS (positive) m/z 1074.83 (M+/2), 716.97 (M+/3)). Biotinylated SML10 peptide (2.2 mg) was reacted with Biotin-Mal (3.0 mg) in DMSO (0.4 mL) at room temperature for 16 h, diluted with water, and lyophilized. Following purification by HPLC and lyophilization, the peptide-Biotin conjugate was verified by ES-MS (ES MS (positive) m/z 973.24 (M+/2), 649.19 (M+/3).

### 2.9. Animals

Polyclonal antibodies were generated in New Zealand White rabbits (CedarLane Laboratories, Burlington, Ontario, Canada). For mouse immunizations, 8- to 10-week-old specific-pathogen-free female C57BL/6J mice were purchased from The Jackson Laboratory (Bar Harbor, ME, USA). The animals were maintained and used in accordance with the recommendations of the Canadian Council on Animal Care Guide to the Care and Use of Experimental Animals. All experimental procedures were approved by the institutional animal care committee (Human Health Therapeutics, National Research Council Canada). For analysis of recurrent infections, 6- to 8-week-old specific pathogen free C57BL/6 mice were purchased from Charles River, UK. Experiments were performed in strict accordance with the requirements of the Animals (Scientific Procedures) Act 1986 and specific procedures were approved by The University of Glasgow Ethical review panel and the U.K. Home Office (project license 60/8797) prior to the initiation of experiments. 

### 2.10. Immunization and Serum Preparation

To produce a polyclonal serum to *C. difficile* spores, formalin killed spores (1 × 10^6^ spores per injection/0.5 mL final volume) emulsified with incomplete Freuds adjuvant (IFA) were used to immunize rabbits subcutaneously. Booster injections of the same formulation (spore numbers in IFA) were given on day 28 and day 56 and immune serum collected day 70 post immunization. This rabbit polyclonal serum was used for ELISA and Western blot analyses. Rabbit polyclonal serum was also raised to the KLH-peptide and KLH-glycopeptide conjugates. Two rabbits were used for each KLH-conjugate antigen and the final serum for each antigen pooled. For the primary immunization, 40 µg of each peptide antigen (50 µL) was mixed with an equal volume of complete Freunds adjuvant (CFA) and delivered by subcutaneous injection. At days 28, 47, and 66 a booster injection with the same amount of antigen mixed with IFA was given by subcutaneous injection. Terminal bleed was on day 78. For peptide/glycopeptide studies in mouse immunization and challenge experiments, the KLH-glycopeptide and KLH-peptide immunogens (5 µg/mouse of peptide) were prepared by mixing with 1 µg/mouse of cholera toxin (CT; Sigma Aldrich, Oakville, ON, Canada) in final volume of 50 µL PBS. The antigen/adjuvant mixture was administered by intranasal inoculation (i.n.) to C57BL/6J mice at day 0 with identical booster immunizations at days 14 and 21. Blood was collected at day 35 for analysis by ELISA. Additionally, some mice were immunized subcutaneously (s.c.) with KLH-glycopeptide admixed with an equal volume of Alum (ThermoFisher Scientific, Ottawa, ON, Canada), as per the above immunization regimen. For *C. difficile* challenge experiments, animals immunized i.n. with CT alone acted as controls to ensure that any response was vaccine specific and not a consequence of a non-specific activation of the immune response by the adjuvant alone. 

### 2.11. SDS-PAGE Analysis and Western Blotting of Spore Extracts 

Spores were resuspended in 100 µL of Laemmli loading buffer and heated to 95 °C for 10 min. Insoluble material was removed by centrifugation and soluble denatured proteins separated using a 12.5% acrylamide gel (Biorad, Hercules, CA, USA). Following electrophoresis, proteins were transferred onto polyvinylidene difluoride membrane (PVDF) and the membrane probed with either rabbit serum generated using formal treated spores (rabbit CD5; 1:50,000), anti-β-O-GlcNAc (Covance, Montreal, ON, Canada; 1:5000) or of KLH-glycopeptide/peptide (raised in rabbits; 1:20,000). Secondary antibodies used included anti-rabbit IgG-HRP conjugate (1:40,000) at or anti mouse IgM HRP conjugate (1:10,000) in PBS-0.1% skimmed milk. Western blots were developed with ECL^TM^ Prime Western blot detection kit according to the manufacturer’s instruction and followed by exposure to X-ray film. 

### 2.12. Evaluation of Vaccine Mediated Protection in the Recurrent Mouse Models of C. difficile Disease 

To determine the impact of Bcl3 specific antibody on spore persistence and transmission, animals were challenged using the recurrent model of *C. difficile* disease. 

In brief, animals were vaccinated i.n. with either KLH-glycopeptide or KLH-peptide admixed with cholera toxin as described above. Nine animals, housed in two independent sterile individual ventilated cages IVC cages and fed sterile food and water ad libitum, were vaccinated with each immunogen and challenged 28 days post final vaccination (Figure 1). To ensure susceptible to infection, animals were given drinking water supplemented with clindamycin (250 mg/L) for 3 days prior to infection. Normal water was replaced for 2 days and then animals were orally challenged with 1 × 10^6^ spores of *C. difficile* R20291. Infection progression was monitored quantitatively through directly plating of diluted fecal material onto Taurocholate Cycloserine-Cefotoxitin Fructose agar (TCCFA) plates and qualitatively though measurement of mouse weight. Two weeks after challenge, animals were treated for 7 days with drinking water containing vancomycin (400 mg/L) (replenished daily) to suppress *C. difficile* vegetative growth. Following removal of this treatment, recurrent disease was determined by measurement of individual mouse weight and recovery of the bacteria with the fecal material (as described above). 

### 2.13. ELISA Assays

ELISA was performed using serum from immunized animals either by coating wells of a microtiter plate (NUNC) with formalin killed spores or streptavidin plates (ThermoFisher, Ottawa, ON, Canada) with biotin–glycopeptide or biotin-peptide antigens. Following blocking with skimmed milk and washing with PBS/Tween solution (×3), pre/immune rabbit and mouse serum was titrated across wells and incubated for 2 h at RT. Following washing with PBS/Tween solution (×3), an IgG or IgA HRP- or AKP-labelled anti-mouse or anti-rabbit secondary antibody was added for 1 h prior to development with horse radish peroxidase HRP or alkaline phosphatase AKP substrate. Titers of antigen specific antibodies are expressed in Figure 5 as the reciprocal of the dilution of serum that generated an optical density (OD) reading 0.5 units (for IgG) and 0.3 OD units (for IgA) above the pre-immune serum from the same animals. 

## 3. Results

### 3.1. Immunogenicity of BclA3 Spore Glycoprotein 

Polyclonal rabbit serum (CD5) to formalin killed R20291 spores was generated in rabbits as described in 2.10. To obtain surface extractable material, spores (1 × 10^6^) were resuspended in SDS-PAGE Laemmli solubilization buffer and heated to 95 ℃ for 10 min. Following removal of insoluble material by centrifugation, the solubilized spore extract was separated on SDS-PAGE gels and transferred to PVDF membrane. Reactivity of the immune serum (CD5) to either R20291 or R20291ΔsgtA spore extracts was assessed by Western blot and revealed that the R20291 spore extract contained two high molecular weight bands of mass >250 kDa, as well as a lower band of approx. mass 100 kDa that reacted with the anti-spore serum (Figure 2, lane 5). In contrast, analysis of R20291Δ*sgtA* spore extracts (Figure 2, lane 6) showed a complete loss of the high molecular weight spore-specific immunoreactive bands in the Western blot. In earlier work [12] we had identified the BclA3 glycoprotein as a component of this high molecular weight extractable material and shown that it reacted with a commercially available *O*-GlcNAc reactive antibody (Covance Inc., Princeton, NJ, USA). In this study we confirmed by Western blotting that the polyclonal spore-specific serum and the *O*-GlcNAc reactive antibody reacted with the same high molecular weight material (Figure 2, lane 1). No reactivity of this CD5 anti-spore polyclonal antiserum was observed when whole cell lysates of vegetative *C. difficile* R20291 cells were examined by Western blotting using this spore specific polyclonal serum confirming that the immunoreactivity observed was to a spore specific antigen(s). 

### 3.2. In Silico Analysis of BclA3 Glycoprotein and Selection of Synthetic Peptides

As multiple efforts to purify the BclA3 protein from the spore surface were unsuccessful, we used previous mass-spectroscopy based glycopeptide identification [12] as well as in silico analysis of the BclA3 protein to select amino acid sequences for peptide synthesis and used these for in vitro glycosylation reactions. The BclA3 protein is a large protein (predicted mass 58.2 kDa) and using TMHMM software [23,24] it is predicted to have at its C terminal (aa 500–678) four transmembrane helices. In addition, the central region of the protein has an extensive collagen-like region which contains multiple glycine-proline-threonine (GPT) amino acid triplet repeats and is predicted to be located on the exterior of a membrane bilayer. We had previously provided evidence to demonstrate that peptides from this region were glycosylated in *O*–linkage at threonine residues. Finally, the N terminal region (aa 1–55) was also predicted to have an exterior location by TMHMM software (http://www.cbs.dtu.dk/services/TMHMM-2.0/) but did not contain the GPT triplet repeated sequence. Peptides selected for in vitro glycosylation reactions are located within the collagen-like region and the N terminus and are listed in Table 1.

### 3.3. Recombinant Expression of SgtA Glycosyltransferase and in vitro Assay Optimization

The *sgtA* gene encodes a protein of 42.1 kDa (358 aa) and contains 10 cysteines. The protein has two distinct domains identified through blast homology search. At the N terminal region of the protein lies the glycosyltransferase GT domain (aa 1–260) which has homology to glycosyltransferase GT-A type superfamily (aa 3–225, E value 7.28e-45) and to the WcaA family of glycosyltransferases (aa 1–255, E value 5.85e-17). At the C terminus of the protein, four tetratricopeptide repeat (TPR) domains were identified (aa 262–358, E value 3.34e-04) with homology to TPR_11/pfam13414. The GT and TPR domains are common to several other well characterized *O*-GlcNAc transferases found in *Xanthamonas campestris* as well as a number of higher eukaryotes [25,26,27]. The TPR repeat is known to mediate protein-protein interactions. 

Recombinant expression of both full length (aa 1–358) and truncated *C. difficile* 630 SgtA (aa 1–227, aa 1–260) using either an N or C terminal His tag was unsuccessful. Successful expression of soluble enzyme was obtained by generating a full-length fusion protein with the MalE protein from *E. coli* [28] and purification of this fusion protein using amylose resin. To examine the activity of the purified MalE-SgtA fusion protein, in vitro assays were set up with synthetic peptides SML1, SML2, and SML3 labelled with FITC-Ahx (Peptide 2.0 Chantilly, VA, USA) (Table 1) using UDP-GlcNAc, UDP-GalNAc, or GDP-GlcNAc sugar nucleotides as donor. Only UDP-GlcNAc was utilized by the MalE-SgtA GT fusion protein as a donor in the in vitro reactions with each of the FITC tagged peptides SML 1, 2, and 3. All in vitro reactions with the three peptides produced products which were glycosylated by the SgtA enzyme with SML3>SML2>SML1 as preferred substrate. The products could be detected by thin layer chromatography (TLC, or capillary electrophoresis (CE) and reaction products obtained with SML3 following 1 and 24 h incubation times are shown in Figure 3. The reaction conditions for the SgtA GT were optimized with the FITC tagged SML3 peptide and were shown to be metal ion dependent (Mn^2+^ > Mg^2+^). All subsequent reactions were performed in 50 mM HEPES buffer pH 7.5, 10 mM MnCl_2_, 0.5 mM DTT, 1 mM UDP-GlcNAc. Multiple reaction products were obtained with each peptide after 24 h incubation which was later confirmed by mass spectrometry MS and NMR to reflect the addition of further GlcNac residues to each peptide at distinct threonine sites (SML1-2T, SML2-3T, SML3-4T). 

Other synthetic FITC tagged peptides based on SML3 peptide sequence were synthesized which contained either a single threonine residue (SML11, SML13, Table 1) or single serine residue (SML14) or where each of the four threonine residues were replaced with serine residues (SML 12). The conversion rates for each FITC tagged peptide after 24 h incubation is presented in Table 2. Glycosylation was observed when threonine residues were replaced by serine (peptides SML12/SML14) indicating the sgtA specificity was not restricted to threonine. Although we recombinantly expressed and tested a number of other putative *C. difficile* glycosyltranferases we were never able to identify any activity which would extend the glycan structure on the peptide backbone in a similar fashion to that found on BclA3 exosporangial protein extracted from the spore surface [12].

To obtain sufficient material for NMR structural analysis of the glycopeptide product a 5 mg batch of a non-FITC tagged BclA3 peptide which had an additional N-terminal cysteine residue (Table 1, SML10) was used in scaled up reactions to produce sufficient material for NMR analysis. The BclA3 glycopeptide SML10 was analyzed by NMR to confirm the addition of a β-*O*-linked GlcNAc monosaccharide residue to the peptide backbone at threonine residues (Table 3). 2D NMR spectra of the glycopeptide contained one spin system of *N*-acetylglucosamine A. However, since the MS analysis of this material showed the presence of peptide with one or two GlcNAc attached, we have to admit that two systems were overlapped, which can be expected. A nuclear overhauser effect (NOE) was observed from GlcNAc H-1 to Thr H-3, indicating glycosylation of the two Thr residues. There were four signals of Thr H/C-3 in HSQC spectrum, two corresponding to non-substituted Thr at 68.4 ppm and two of the glycosylated Thr at 76.2 ppm (Table 3). We could not assign the positions of the two glycosylated Thr in the peptide chain by NMR.

### 3.4. Conjugation of BclA3 Peptide and BclA3 Glycopeptide to KLH

To obtain sufficient quantity of glycopeptide for production of a conjugate we scaled up the glycosylation reaction with the purified MalE-SgtA fusion protein. A 10 mg batch of SML10 peptide (peptide SML3 with N terminal cysteine added) was glycosylated as described in Section 2.7. Following 24 h incubation the reaction was stopped by flash freezing and the products separated on a Luna C18 reverse phase column with water-acetonitrile gradient containing 0.1% TFA (0%–70% in 25 min, flowrate of at 4 mL/min). Although multiple peaks were observed on initial C18 separation, the sample was resolved into a single, major peak and two minor peaks following TCEP treatment to reduce any disulfide bond (10 min at room temperature). Electrospray mass spectrometry (ES-MS) analysis of the major peak confirmed that this fraction contained only peptide with a single GlcNAc residue attached. MS analysis of fractions containing the minor peaks confirmed that these fractions contained SML10 peptide with two HexNAc residues attached. 

HPLC purified SML10 glycopeptide with a single HexNAc residue or non-glycosylated SML10 peptide were then used for in conjugation reactions with KLH or BSA as described in Section 2.7. BSA conjugates were analyzed by matrix assisted laser desorption/ionization (MALDI) mass analysis and determined to contain 12%–14% (w/w) of peptide antigens (approx. estimate ratio of peptide-BSA, 9.5:1; glycopeptide-BSA, 8.5:1). The antigen content in KLH-conjugates was estimated to be equivalent based on the BSA conjugates which were made in parallel. 

### 3.5. Evaluation of Peptide Antigenicity

To determine if the BclA3 peptide or glycopeptide were recognized by polyclonal serum raised to either formalin-killed spores or formalin-killed vegetative cells we used an ELISA assay (Figure 4). BSA conjugated peptides SML4, SML7, SML8, and SML10 and BSA conjugated SML10 glycopeptide were coated on ELISA plates. Titrations of rabbit polyclonal serum CD5 (anti-spore) and CD2 (anti-vegetative cells) revealed that only the BclA3 SML10 glycopeptide was recognized by CD5 anti-spore serum while no non-glycosylated peptide from either the collagen repeat region (SML10, SML4) or the surface exposed N terminal region (SML7, SML8) of the protein was reactive with this polyclonal serum. As expected, none of the BclA3 peptides or BclA3 glycopeptide were recognized by CD2 anti-vegetative cell serum.

### 3.6. KLH-Conjugate Animal Immunization

Mice and rabbits were immunized with either KLH-glycopeptide or KLH-peptide conjugates as described in Section 2.9. Immune serum was collected and analyzed by ELISA using either biotinylated glycopeptide, biotinylated peptide, or formalin-killed spores as coating antigen. The response of immunized rabbits to the KLH- peptide or KLH-glycopeptide admixed with Freunds adjuvant and delivered subcutaneously is presented in Figure 5. While immunization of rabbits with the KLH-glycopeptide conjugate produced an IgG response which could recognize both peptide and glycopeptide antigens, immunization with the KLH-peptide conjugate produced an IgG response which recognized the non-glycosylated peptide but only had weak reactivity towards the glycopeptide (Figure 5A,B). Significantly, only the KLH-glycopeptide immune antiserum was capable of recognizing the natural BclA3 glycoprotein on the spore surface as demonstrated by ELISA using *C. difficile* R20291 spores as coating antigen (Figure 5E). With respect to serum IgA response in rabbits to either KLH-peptide or KLH-glycopeptide conjugates, only a weaker measurable IgA titer to the glycopeptide antigen was observed following immunization with KLH-glycopeptide conjugate (Figure 5C,D). Western blot analysis using R20291 and R20291::sgtA spore extracts separated by SDS-PAGE, were incubated with KLH-SML10 glycopeptide or KLH-SML10 peptide antiserum. Only the KLH-glycopeptide serum reacted with R20291 spore extract material (Figure 2, lane 3). This reactivity was to the same high molecular weight material recognized by both the CD5 (anti-formalin killed spore serum) and anti–*β O*-linked GlcNAc specific antibody (Figure 2, lanes 1 and 5). As would be expected, spore extracts from R20291::sgtA mutant strain were no longer reactive with the KLH-glycopeptide antiserum (Figure 2, lane 4). No reactivity to spore extract material was observed when antiserum raised to KLH- SML10 peptide was tested.

For immunogenicity testing in mice we used the KLH-glycopeptide conjugate and compared intranasal delivery with cholera toxin versus subcutaneous delivery route using alum as an adjuvant. Serum IgG responses to the KLH-glycopeptide or KLH-peptide conjugate were measured by ELISA and are presented in Figure 6. As with the rabbit immunization experiment, mice immunized with KLH-glycopeptide intranasally (i.n.) with CT recognized both peptide and glycopeptide antigens although a much higher titer to the glycopeptide was produced (Figure 6A,B). Subcutaneous (s.c.) immunization of KLH-glycopeptide admixed with alum produced only a weak serum IgG response to BclA3 glycopeptide and no detectable IgG response to BclA3 peptide as measured by ELISA.

### 3.7. Recurrent Mouse Model

To determine whether antibodies raised to KLH-glycopeptide could influence persistence of *C. difficile* spores within the mouse model of acute and relapsing disease, mice were immunized intranasally with either KLH-glycopeptide, KLH peptide, or KLH alone admixed with CT to act as a mucosal adjuvant. Serum samples collected post three immunizations confirmed earlier observations that animals immunized with KLH-glycopeptide generated higher titers to the BclA3 glycopeptide than the peptide alone. In contrast titers of antibodies were higher to the peptide when the KLH-peptide was used as the immunogen (Figure 7A). Interestingly low levels of IgA specific antibodies to both antigens were only detected following immunization with KLH-glycopeptide (Figure 7B). 

Following *C. difficile* challenge, all animals lost weight on day 2, indicative of acute disease (Figure 8A). Fecal samples collected from these animals on day 2 and on subsequent days (up to 14 days post challenge) confirmed animals were colonized with *C. difficile*, with no differences in either total (Figure 8B) or spore counts (Figure 8C) observed between animals vaccinated with KLH-glycopeptide, KLH-peptide, or KLH alone. To determine whether antibodies to KLH-glycopeptides limit relapse, animals were treated for 7 days with vancomycin; currently a standard treatment for severe disease in humans. This reduced *C. difficile* colonization to undetectable levels by culture. On removal of this treatment, animals were observed to relapse, with consequential weight loss observed. No difference in extent of relapse was observed, with equivalent numbers of bacteria recovered from the tissues of animals independent of vaccination formulation formulations. These data indicate that at least in the context of these experiments, vaccination with the BclA3 glycopeptide does not enhance protection against *C. difficile* disease. 

## 4. Discussion

The severity of disease and associated mortality of recurrent infections resulting from the emergence of new hypervirulent *C. difficile* strains has spurred research into finding alternative measures to prevent infection or to reduce rates of recurrence. Effective vaccine therapy offers the opportunity to provide long term protection against future CDI episodes. To date, most effort has been directed against neutralizing the two key virulence factors TcdA and TcdB [29,30,31] through targeting of these antigens by vaccination or via passive immunization strategies [16]. However, this approach does not prevent colonization by the organism, leading to asymptomatic carriage of *C. difficile*. Patients infected with *C. difficile* can shed high levels of dormant spores which are the primary routes of dissemination and spread of infection and as such must be included in any comprehensive, therapeutic treatment plan. An ideal prophylactic strategy should target both toxins (to prevent intestinal damage) and spores (to reduce the transmission and carriage of this critical infectious entity). 

The success of carbohydrate-based subunit vaccines for treatment of other important bacterial infections [32,33,34] was an important factor which led to our study of the glycan structures produced by both vegetative cells and spores of *C. difficile* as potential subunit vaccine candidates [12,35]. Although spores are a metabolically dormant form of *C. difficile*, which persist in the gut during antibiotic treatment, recent studies have indicated that the exosporangial layer may be involved in host cell interactions and as such may present a good therapeutic target [19,36]. Hair-like projections similar to those found on *B. anthracis* spore surface have been identified on *C. difficile* spores. Studies of these structures on *B. anthracis* spores indicate that they contain the collagen-like exosporium BclA and BclB glycoproteins [37]. In a similar fashion, the Bcl collagen-like proteins BclA1, BclA2, and BclA3 of *C. difficile* have been shown to localize exclusively to the exosporium of *C. difficile* spores [38] and in an earlier study we demonstrated that the BclA3 protein was glycosylated in a novel fashion. A previous effort to determine the utility of vaccination with the collagen-like BclA proteins of *C. difficile* used recombinant (non-glycosylated) BclA1 administered intraperitoneally to mice and provided no detectable protection upon challenge [39]. However, limitations of that study included the IP administration of the recombinant protein antigen which may restrict immunity to a humoral response and the selection of recombinant non-glycosylated BclA1 protein as the vaccine antigen. Additionally, a number of isolates including the UK1 challenge strain used in the above studies appear to express a truncated version of the BclA1 protein. As such the recombinant BclA1 antigen used for vaccination trials may explain the lack of efficacy in the mouse challenge. 

In the current study we focused on the BclA3 protein which is common to most *C. difficile* strains and so may provide broader protective immunity towards a large number of clinical isolates. The contribution of the extensive glycosylation in the collagen repeat region of the BclA3 protein towards protein antigenicity has not been examined to date. In this study we show that immune serum raised to formalin killed *C. difficile* spores preferentially recognized a BclA3 glycopeptide when compared to the same BclA3 peptide lacking glycosylation and only immune serum raised to KLH-conjugated BclA3 glycopeptide (and not KLH-BclA3 peptide) was able to react with the Clostridial spore surface demonstrating the significance of glycan moieties on the immunogenicity of this key exosporangial protein. As extraction and purification of the BclA3 glycoprotein from the spore surface was challenging we used synthetic BclA3 peptides and produced glycopeptides by in vitro reactions using a recombinant form of SgtA, an *O*-GlcNAc transferase. The *sgtA* gene is located immediately upstream of the *bclA3* gene in the *C. difficile* genome. In this manner we were able to demonstrate that the SgtA glycosyltransferase was responsible for the addition of the initiating β-*O*-linked GlcNAc moiety to individual threonine residues within a number of peptides from the collagen repeat region of the BclA3 protein. It is important to note, however, that although we recombinantly expressed and tested a number of other putative *C. difficile* glycosyltranferases in in vitro assays with peptide/glycopeptide we were never able to identify any additional activity which would extend the glycan structure on the peptide backbone to that found on BclA3 exosporangial protein extracted from the spore surface [14]. As such we were limited to using a BclA3 peptide glycosylated with a single GlcNAc monosaccharide added to a threonine residue as a BclA3 peptide subunit vaccine candidate antigen in this study. While we demonstrated that a specific serum IgG response could be generated to either the glycopeptide or peptide following conjugation of each peptide to KLH carrier protein, only antibodies produced to the glycopeptide were able to bind to the spore surface. This suggests that glycan modification of multiple threonine residues from this surface exposed CLR region is readily accessible on spore surface and likely masks underlying peptide epitopes. 

As anticipated, i.n. immunization of mice with CT as mucosal adjuvant induced antigen-specific IgA and IgG responses in the serum whereas s.c. immunization of mice with alum elicited only antigen-specific IgG responses. Since *C. difficile* colonizes the large intestinal surface and the intestine is recognized as the initial site of infection, it is postulated that antigen specific IgA may play a role in the protection against the colonization of *C. difficile* in the intestine. However, within these studies, we did not determine if antigen specific IgA was present in the gut at time of challenge. Challenge studies, using both acute and relapsing models of disease in mice showed that vaccination did not influence colonization or persistence. The failure of this approach suggests that either immunity to the glycosylated BclA3 protein is not protective, however, it could also indicate that presentation of the antigen is insufficient to stimulate sufficient levels of immunity to drive protection. Recently, Broecker, et al. [40] and Monteiro [41] reported protection against *C. difficile* was achievable by vaccination using glycans of the surface polysaccharides of *C. difficile*, PS-I, PS-II, and PS-III. Surprisingly, all three offered full protection against colonization and pathology, despite the fact that expression of these antigens appears to vary between strains in vitro. This may indicate that future optimization of the vaccine formulation and the use of alternative routes of immunization, such as oral delivery to the gut mucosa to stimulate immunity at site of infection, may improve the quality of the induced immune responses and thereby provide better immune protection. 

In addition, although antibodies raised to the KLH-BclA3 glycopeptide which expressed a single GlcNAc moiety were able to react with the spore surface, as demonstrated by ELISA, this may not be the optimal spore surface glycopeptide antigen. While previous structural characterization of the BclA3 glycoprotein indicated that a single HexNAc monosaccharide was present at some sites within the collagen repeat region of the BclA3 protein, a more complex glycan structure with a unique capping moiety was the predominant structure present in this collagen repeat region. As such a more robust antibody response to this extended glycan may provide better immune protection and/or spore decolonization. Future studies will be directed towards identifying additional spore glycan biosynthetic enzymes to facilitate the generation of the extended spore glycan. This will also provide material to complete a detailed structural analysis of the extended BclA3 glycan and provide material for conjugation/vaccine studies. 

## 5. Conclusions

It is clear from this study that vaccination with antigens that target the spore surface alone are insufficient to protect against *C. difficile* morbidity and mortality and that optimal protection will require the development of a vaccine that combines antigens that target both toxin and cell surface structures.

## Figures and Tables

**Figure 1 vaccines-08-00073-f001:**
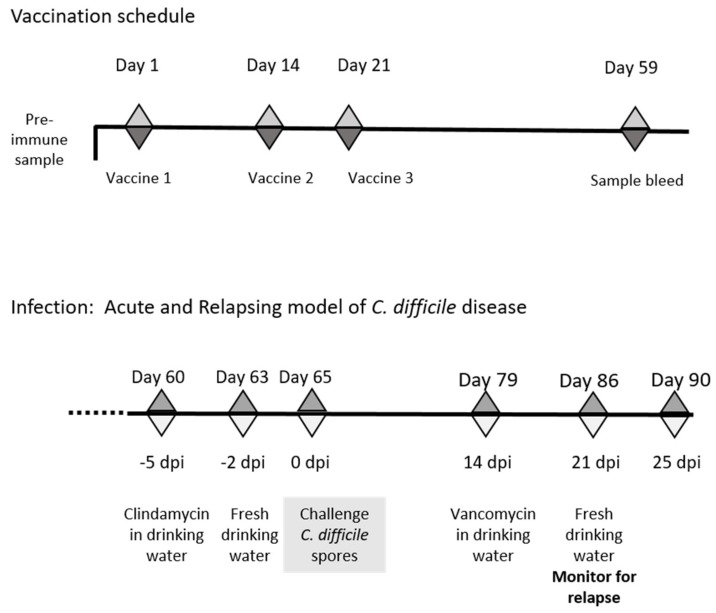
Schematic of schedule of vaccination and antibiotic treatments used to establish acute and relapsing *C. difficile* disease in mice. The vaccination schedule highlights the days vaccines were given and sample bleeds taken for immune evaluation, whilst the infection schedule indicates the timing of antibiotic treatments and *C. difficile* challenge.

**Figure 2 vaccines-08-00073-f002:**
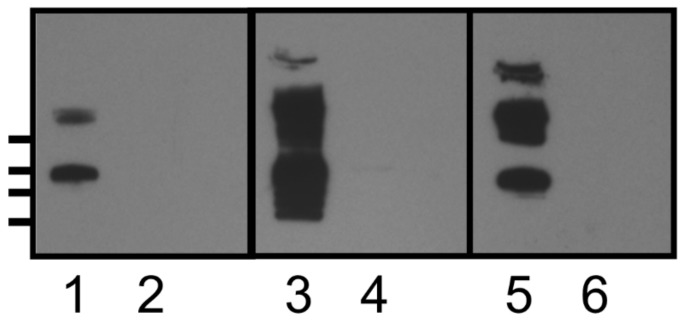
Western blot analysis of spore surface extracts. Lanes 1, 3, and 5—*Clostridium difficile* R20291 spore surface extract; lanes 2, 4, and 6—R20291::sgtA spore surface extract. Lanes 1 and 2 reacted with anti-β-*O*-linked GlcNAc antibody, lanes 3 and 4 reacted with rabbit polyclonal anti KLH-SML10 glycopeptide antiserum, lanes 5 and 6 reacted with rabbit anti-formalin killed spore antiserum (CD5). Molecular weight markers indicated on left hand side (top to bottom) 250, 150, 100, and 75 kDa.

**Figure 3 vaccines-08-00073-f003:**
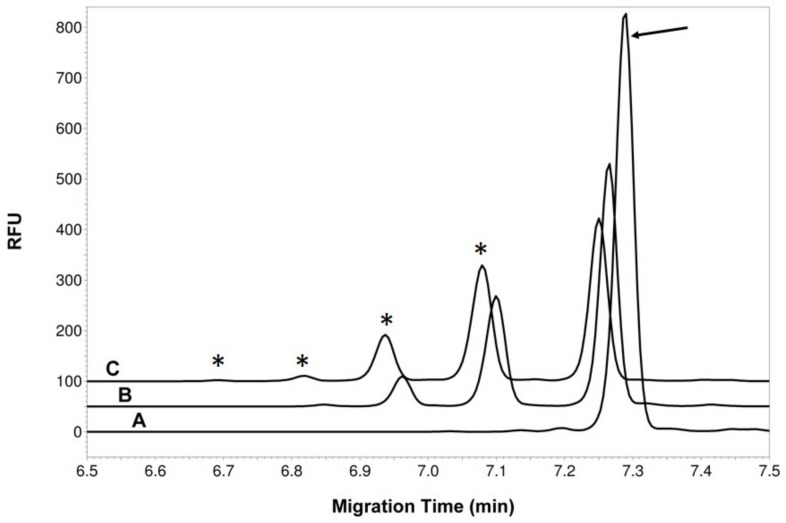
Capillary electrophoresis (CE) analysis of the MalE-SgtA reaction products. Electropherogram **A** is the acceptor alone (SML3 peptide labeled with FITC-Ahx, peak indicated by arrow) incubated for 24 h in presence of MalE-SgtA but without donor. Electropherogram **B** shows the complete reaction mix after 1 h incubation. Electropherogram **C** shows the complete reaction mix after 24 h incubation. The “*” indicates product peaks with added GlcNAc residue. The traces are shifted by 0.2 min to facilitate viewing of the peaks.

**Figure 4 vaccines-08-00073-f004:**
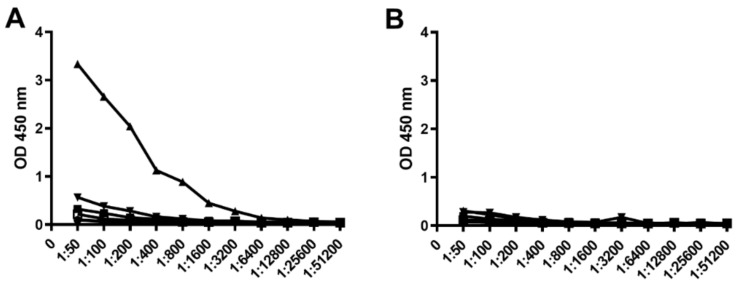
ELISA of BSA conjugated peptide/glycopeptide antigens with *C. difficile* anti-spore (**A**) and *C. difficile* anti-vegetative cell (**B**) polyclonal serum. Coating antigens are indicated as follows: BSA-SML10 peptide (black square), BSA-SML10 glycopeptide (regular triangle), BSA-SML4 peptide (inverted triangle), BSA-SML7 peptide (rhombus), BSA-SML8 peptide (circle), BSA (white square).

**Figure 5 vaccines-08-00073-f005:**
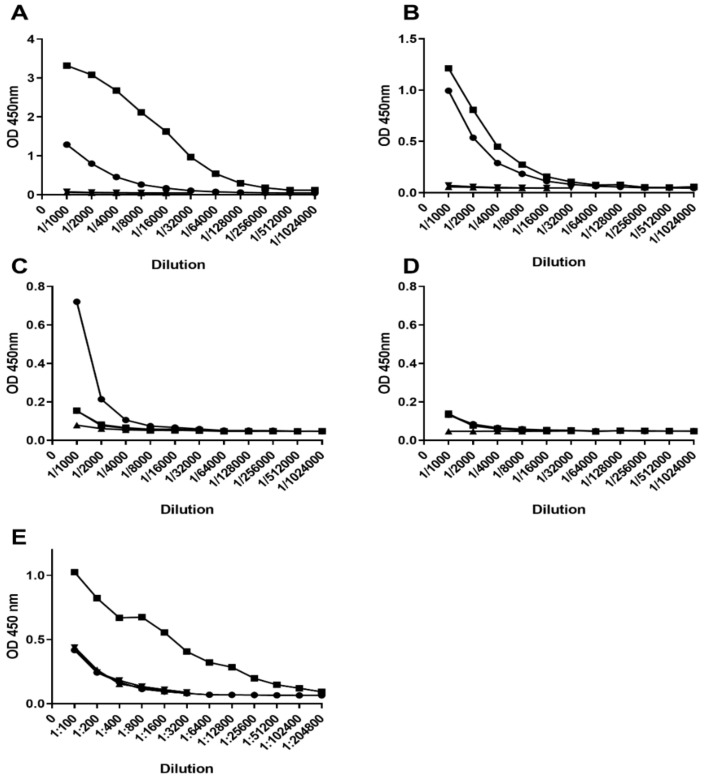
ELISA reactivity of polyclonal rabbit serum raised to KLH-conjugated peptide or KLH-conjugated glycopeptide. Panels (**A**,**C)** coating antigen biotin-SML10 glycopeptide, panels (**B**,**D**) coating antigen biotin-SML10 peptide, Panel (**E**) coating antigen formalin killed *C. difficile* R20291 spores. Panel (**A**,**B**,**E**) serum IgG response and panels (**C**,**D**) serum IgA response. Legend: KLH-glycopeptide immune serum titration (black square), (KLH-peptide immune serum titration (circle), KLH-glycopeptide pre-immune serum titration (regular triangle), KLH-peptide pre-immune serum titration (inverted triangle).

**Figure 6 vaccines-08-00073-f006:**
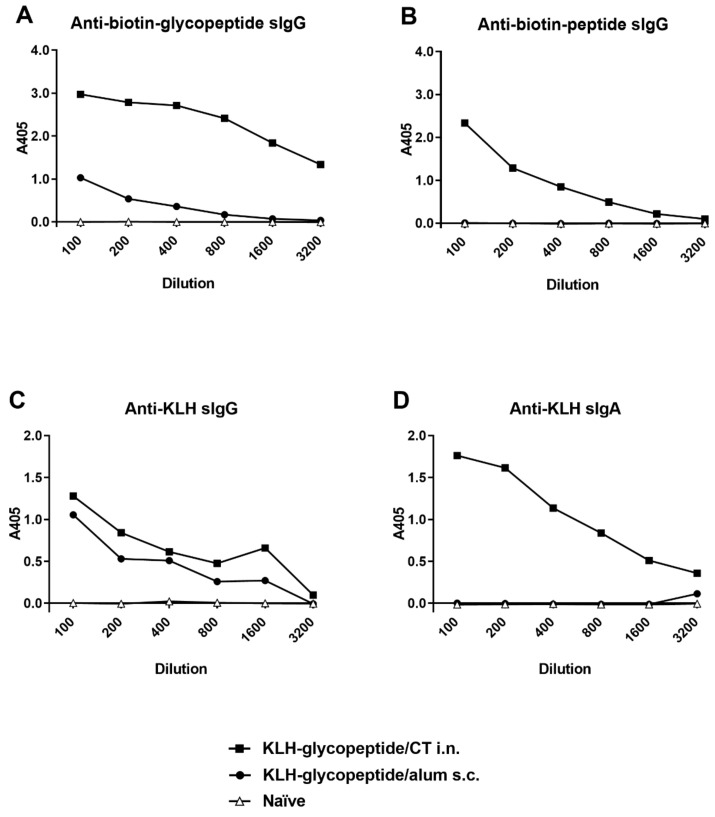
Mouse immune response to KLH-SML10 glycopeptide—comparison of immunization route and adjuvant. (**A**). Serum IgG recognition of Biotin-SML10 glycopeptide coating antigen. (**B**). Serum IgG recognition of Biotin-SML10 peptide coating antigen. (**C**). Serum IgG recognition of KLH. (**D**). Serum IgA recognition of KLH.

**Figure 7 vaccines-08-00073-f007:**
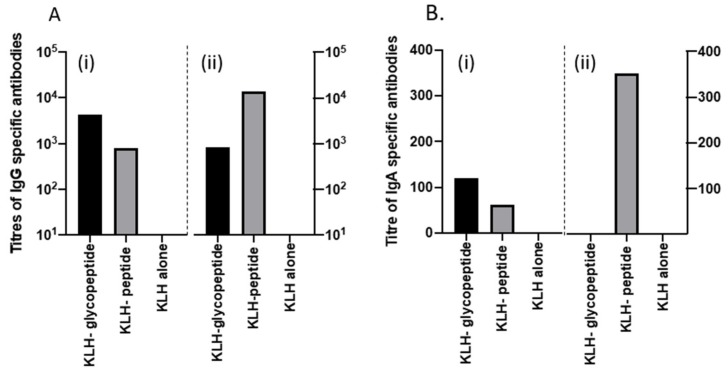
Antibody response to intranasal vaccination of the peptides and CT in the serum of mice following immunizations on days 1, 14, and 21. (**A**) shows the IgG specific response to both the KLH glycopeptide conjugate (balck), KLH peptide (grey), and KLH alone (white), respectively. Figure 7A (i) shows the calculated titers when the glycopeptide was used as the coating antigen and 7A (ii) when the peptide alone was used. (**B**) shows the IgA specific response to both the KLH glycopeptide conjugate (black), KLH peptide (grey), and KLH alone (white), respectively. Figure 7B (i) shows the calculated titers when the glycopeptide was used as the coating antigen and 7B (ii) when the peptide alone was used.

**Figure 8 vaccines-08-00073-f008:**
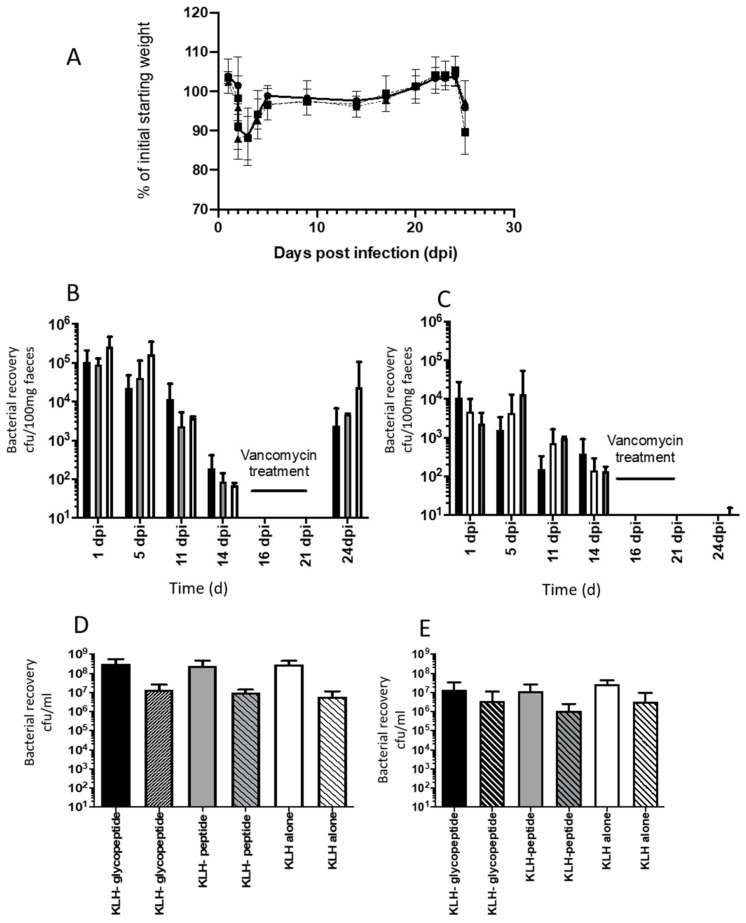
Impact of intranasal vaccination of mice with KLH peptide conjugates + CT on acute and relapsing *C. difficile* disease. (**A**) represents changes in average percentage (%) change in the weight of the mice following *C. difficile* challenge. Each point presents the average weight of animals immunized with either KLH-glycopeptide + CT (circle), KLH-peptide + CT (inverted triangle), and KLH + CT alone (black square). Error bars represent the standard deviation of the mean for 8–9 mice per group. (**B**) reflects the recovery of both vegetative and spores in the feces of mice on days post challenge with *C. difficile* (days post infection, dpi). The mean bacterial recovery (cfu/100g of feces) from animals immunized with KLH-glycopeptide + CT (black square), KLH- peptide + CT (grey square), and KLH + CT (white square) from 8–9 mice per group is shown. Error bars represent the standard deviation from the mean. (**C**) represents analysis of the spores within the samples highlighted in Figure 6B and were calculated by from serial dilutions of samples following treatment of these samples at 65 °C for 30 min. (**D**) represents the recovery of *C. difficile* from caecal samples collected at the experimental endpoint on day 25. Mean bacterial total counts from 8-9 animals immunized intranasally with KLH-glycopeptide + CT (black square), KLH-peptide + CT (grey square), or KLH + CT (white square) are shown in single blocked colors and spore counts from the same groups are highlighted by patterned blocks. Error bars represent the standard deviation from the mean of these counts. (**E**) represents the equivalent recovery of *C. difficile* from colonic samples from the same animals, with total recovery of *C. difficile* represented by solid blocks of color and spore counts by patterned blocks. Error bars represent the standard deviation from the mean of these counts.

**Table 1 vaccines-08-00073-t001:** Synthetic BclA3 peptides.

Peptide Name	Sequence (R20291 BclA3)	Tag/Conjugate
(BclA3 aa Sequence)		
SML1 (308–319) 12 aa	AGLIGPTGATGV	FITC-Ahx
SML2 (391–404) 15 aa	VGPTGATGATGADGV	FITC-Ahx
SML3 (278–292) 15 aa	TGATGLIGPTGATGA	FITC-Ahx
SML4 (391–404) 15 aa	VGPTGATGATGADGV	KLH and BSA conjugate
SML9 (271–295) 25 aa	GATGIGITGATGLIGPTGATGATGA	FITC-Ahx
SML10 (278–292) 16 aa	(C)TGATGLIGPTGATGA	N terminal cysteine/KLH and BSA conjugate
SML11 single site	DGAVGLIGPTGAGADGA	FITC-Ahx
SML12 T to S, 4 sites	SGASGLIGPSGAGASGA	FITC-Ahx
SML13 single T site	DGATGLIGPDGAGADGA	FITC-Ahx
SML14 single S site	DGASGLIGPDGAGADGA	FITC-Ahx
SML7 (1–15) 15aa	(C)MSRNKYFGPFDDNDYN	N terminal cysteine/BSA conjugate
SML8 (46–60) 15aa	(C)VGPTGPMGPRGRTGP	N terminal cysteine/ BSA conjugate

**Table 2 vaccines-08-00073-t002:** BclA3 peptide glycosylation conversion rates of reactions with MalE-SgtA and UDP-GlcNAc. The % conversion rates correspond to the integration of all the product peaks identified by CE analysis. #: number of potential sites.

Peptide (T/S, # of Potential Sites)	% Conversion (1 h)	% Conversion (24 h)
SML1 (T,2)	13.0	17.0
SML2 (T,3)	21.5	45.5
SML3 (T,4)	38.9	52.6
SML11 (T,1)	17.1	28.1
SML12 (S,4)	10.0	46.6
SML13 (T,1)	24.1	39.0
SML14 (S,1)	40.6	70.6

**Table 3 vaccines-08-00073-t003:** Purified SML10 glycopeptide reaction product analysis. NMR data (600 mHz, 25 ℃) NAc: 2.05/23.4 ppm. Thr ^x^ and Thr ^y^ are glycosylated with β-GlcNAc. “x” and “y” are used to describe two distinct threonine residues.

	H/C-1	H/C-2	H/C-3	H/C-4	H/C-5	H/C-6
β-GlcNAc A	4.56	3.69	3.55	3.45	3.44	3.76; 3.90
	101.0	56.6	74.8	71.0	77.0	61.9
Thr ^x^		4.52	4.31	1.18		
		58.7	75.9	17.1		
Thr ^y^		4.54	4.34	1.18		
		58.7	75.9	17.1

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
