# Peer review of "In Vitro Production and Immunogenicity of a Clostridium difficile Spore-Specific BclA3 Glycopeptide Conjugate Vaccine"

_vaccines, 2020, doi:10.3390/vaccines8010073_

Round 1
Reviewer 1 Report
The article is devoted to the urgent problem of creating a vaccine against the spore-forming bacterium Clostridium difficile. A feature of this disease is the resistance of the spores to antibiotics. To date, there are no vaccines that cause an immune response to the immunodeterminant groups of the bacterial spore membrane.
The role of hexosamine transferase in the biosynthesis of spore glycoconjugates is convincingly proved in the article. On the other hand, the authors obtained a recombinant enzyme. A glycopeptide that mimics spore antigens was synthesized with its help. Based on it, the immunogen was synthesized and its effectiveness has been proven. Antibodies to it actively interact with spore glycoproteins. Unfortunately, intranasal immunization of mice did not protect them from a lethal dose of bacterial spore. Perhaps this is due to the low efficiency of intranasal immunization. Perhaps a more complex glycan structure is required. In any case, synthetic vaccines have low side effects compared to natural antigens. Standardization of such vaccines is much simpler. The article will be interesting to many readers.
The main advantage of this article is an attempt to develop a vaccine against spores of this bacterium. The ineffectiveness of antibiotic therapy against bacterial spores is a serious problem that needs to be addressed. But this approach being developed is promising and may lead to brilliant results in the future. The results obtained at this stage will be interesting and useful to readers. I believe that the article can be accepted for publication.
Strengths of the manuscript.
The critical role of transferase in the synthesis of immunodominant groups of spore surface has been proved. Bacteria knocked out by this enzyme are not recognized by antibodies to native spore antigens.
The developed vaccine for intranasal immunization was of little effect and did not protect animals. This is most likely due to the low efficiency of intranasal immunization itself, as well as to the simple structure of the glycopeptide in the immunogen. Antibodies to glycopeptide actively react with surface antigens of spores and BclA3 glycopeptide, which is convincingly proven.
It is possible to make a remark to the authors who use conjugates with BSA (Figure 3) and do not describe their preparation in the methods. But the authors use a similar methodology to obtain immunogens in this manuscript.
Author Response
We would like to thank both referees for the positive response to our submitted manuscript entitled ‘In vitro production and immunogenicity of a Clostridium difficile spore-specific BclA3 glycopeptide conjugate vaccine. We have modified the manuscript in line with the referee’s comments to improve clarity of the approach taken in generation of the conjugates and in their analysis.
Reviewer 1
In response to comments from referee 1, who wished to know ‘whether BSA conjugates were made using the same method as the KLH conjugates’ , we have modified the description of the methodology to clarify that both bioconjugates were generated using the same approach. Changes are highlighted in bold. We hope that this description is now much clearer.
Line 137. Both the BSA and the KLH were made in parallel. Each peptide sample was mixed with an equivalent amount of bromoacetylated KLH (KLH-Br) (5 mg, 2.5 ml), or, BSA-Br (0.8 mg, 0.4 ml) in PBS. The solution was kept at room temperature for 72 h. Purification of the conjugates was achieved using a 30K Centriprep or Amicon at 3000 rpm for 10 min (washed with PBS x 5) to remove free peptides and other reagents. MALDI-MS analysis of respective BSA-conjugated peptide was used to estimate the antigen loading on the KLH conjugate, prepared in parallel, at 12-14% (w/w).
Reviewer 2 Report
Comments to authors
The authors have explored a method using both BclA3-glycopeptide and Bcl3-peptide conjugation with KLH to study the immune response. They demonstrated that the glycan moiety is a predominant spore-associated antigen. Specific antibodies were raised to both antigens. Unfortunately, immunization did not show any protection against acute or recurrent disease. But this approach did provide an alternative method to study the specific disease in the future. There this article is benefit for medicinal chemists or immunologists. Therefore, it is worthy of publication.
One suggestion:
Although both antigen show immune response, but it is good to know the ratio between BclA3-Glycopeptide: KLH. Too dense or too diluted to be an efficient vaccine.
Author Response
We would like to thank both referees for the positive response to our submitted manuscript entitled ‘In vitro production and immunogenicity of a Clostridium difficile spore-specific BclA3 glycopeptide conjugate vaccine. We have modified the manuscript in line with the referee’s comments to improve clarity of the approach taken in generation of the conjugates and in their analysis.
Reviewer 2
In response to comments from reviewer 2 who wished to know the ratio of glycopeptide/peptide to KLH protein, we have calcalculated the following rations which are now included in the text.
Line 343. HPLC purified SML10 glycopeptide with a single HexNAc residue or non-glycosylated SML10 peptide were then used for in conjugation reactions with KLH or BSA as described in Materials and Methods. BSA conjugates were analysed by MALDI mass analysis and determined to contain 12-14% (w/w) of peptide antigens (approx. ratio of Peptide-BSA, 9:5:1; glycopeptide-BSA, 8:5:1). The antigen content in KLH-conjugates was estimated to be equivalent based on the BSA conjugates which were made in parallel.